

# Comparing effects of attractive interactions in crowded systems: nonspecific, hydrophobic, and hydrogen bond interactions

Saman Bazmi and Stefan Wallin

Department of Physics and Physical Oceanography, Memorial University of Newfoundland, St John's, NL, Canada

## ABSTRACT

The equilibrium stability of a protein is determined by its amino acid sequence and the solution conditions, such as temperature, pH and presence of chemical denaturant. The stability of a single protein in two identical solutions can nonetheless differ if other macromolecules, termed cosolutes or crowders, are present in one of the solutions at concentrations high enough to occupy a substantial fraction of the solution volume. This effect, due to the presence of the crowders, decreases or increases the stability depending on the interactions between the protein and crowders. Hard-core steric repulsions, which are responsible for the reduction in free volume, are expected to entropically stabilize the protein while attractive interactions can be destabilizing. Here we use a coarse-grained protein model to assess the impact of different types of crowder-protein interactions on the stability of a 35-amino acid model sequence folding into a helical bundle. We find that, for the same interaction strength and concentration, spherical crowders with a hydrophobic character are more destabilizing than crowders interacting nonspecifically with the protein. However, the two types of interactions differ in the degree of association between crowders and protein. At an interaction strength for which the attractive interactions roughly counteracts the stabilizing hard-core repulsions, the nonspecific interactions lead to much stronger crowder-protein association than the hydrophobic interactions. Additionally, we study crowders in the form of polypeptide chains, which are capable of hydrogen bonding with the protein. These peptide crowders have a destabilizing effect even at relatively low crowder concentrations, especially if the sequence of the peptide crowders includes hydrophobic amino acids. Our findings emphasize the importance of the interplay between different types of attractive crowder-protein interactions and entropic effects in determining the net effect on protein stability.

# INTRODUCTION

Most biophysical experiments on biomolecules or biomolecular systems are performed under dilute solution conditions in which the macromolecular concentration rarely

Corresponding author
Stefan Wallin, swallin@mun.ca

exceeds 10 g/L (*Theillet et al., 2014*). However, the cellular environment is often anything but dilute (*Ellis, 2001*). For example, the concentration of macromolecules in *Escherichia coli* can reach up to 300–400 g/L, corresponding to a volume occupancy of around 30–40% (*Zimmerman & Trach, 1991*). The crowded cellular milieu has been shown to impact a wide range of biophysical processes, including diffusion (*Słyk, Skóra & Kondrat, 2022*; *Wang et al., 2012a*), aggregation (*Schreck, Bridstrup & Yuan, 2020*; *Siddiqui & Naeem, 2018*), DNA replication (*Akabayov et al., 2013*), protein fold switching (*Zhang et al., 2023*; *Bazmi, Seifi & Wallin, 2023*), and liquid-liquid phase separation (*André, Yewdall & Spruijt, 2023*; *Julius et al., 2019*).

One of the major issues in macromolecular crowding, in fact since the inception of the field (*Minton, 1980*), is the effect of crowding on the equilibrium stability of proteins (*Pastore & Temussi, 2022*; *Speer et al., 2022*; *Zhou, Rivas & Minton, 2008*). In the simplest scenario, the protein populates mainly two states (*Sosnick & Barrick, 2011*): the structurally coherent native state, N, and the typically more extended and structurally diverse unfolded state, U. Protein stability reflects the relative population of these two states. It is therefore expected that different types of crowder-protein interactions differently impact stability. Hard-core steric repulsions, which arise simply from the fact that two macromolecules cannot simultaneously occupy the same region in space, are expected to stabilize proteins, because conformations of the extended U ensemble become entropically disfavored relative to the compact N under crowded conditions. Experimental studies employing artificial polymer macromolecules as crowder agents typically (*Sasahara, McPhie & Minton, 2003*; *Spencer et al., 2005*; *Wang et al., 2012b*; *Hong & Gierasch, 2010*; *Christiansen & Wittung-Stafshede, 2014*), although not always (*Nasreen et al., 2018*; *Malik et al., 2012*), stabilize proteins as measured by, *e.g.*, the folding midpoint temperature or the free energy of folding. It was shown, within a coarse-grained model for folding, that volume exclusion by purely repulsive crowders can have a neutral effect on stability, or even lead to a net destabilization, under some conditions (*Bazmi & Wallin, 2022*).

In addition to steric repulsions, attractive or repulsive interactions between the protein and crowders will in general also impact protein stability. Such interactions, often called soft or "chemical", arise from various effects, including charge-charge, van der Waals, hydrogen bonding, and hydrophobic interactions (*Sarkar, Li & Pielak, 2013a*). Soft interactions between protein and crowders are often assumed to be nonspecific (*Sarkar, Li & Pielak, 2013a*; *Guin & Gruebele, 2019*; *Rivas & Minton, 2022*) in contrast to the specific (*Jones & Thornton, 1996*) and cooperative (*Hsu, Yen & Yeang, 2022*) interactions that occur between the components of functionally relevant biomolecular complexes. Repulsive soft interactions are expected to enhance the stabilizing effect of hard-core repulsions, *i.e.*, make the protein even more stable. By contrast, attractive soft interactions between protein and crowders should counteract the hard-core stabilization (*Sarkar, Smith & Pielak, 2013b*). The reason is that, while the conformations of both U and N will make favorable contacts with the crowders, the larger surface area of the U should lead to a net energetic stabilization of this state, and hence provide a destabilizing effect on the protein. Both computational (*Feig & Sugita, 2012*; *Rosen, Kim & Mittal, 2011*; *Kim & Mittal, 2013*; *Douglas, Dudowicz & Freed, 2009*; *Bille, Mohanty & Irbäck, 2016*; *Tsao &*

*Dokholyan, 2010*; *Macdonald et al., 2016*) and experimental (*Phillip, Kiss & Schreiber, 2012*; *Jiao et al., 2010*; *Miklos et al., 2010*) studies have been performed to study the effects of attractive crowder-protein interactions. In general, the net effect on the stability will be determined by a competition between stabilizing steric repulsions and destabilizing soft attractive interactions. Because soft interactions have an energetic (enthalpic) component, the impact of crowders should be temperature-dependent (*Miklos et al., 2010*). Indeed, this was observed by *Zhou (2004)* who studied the stability of the protein CI2 in the presence of protein crowders. CI2 was found to be destabilized at low temperatures and stabilized at high temperatures. This change allowed the authors to define a so-called crossover temperature at which there was no net effect on the stability of CI2 (*Timr & Sterpone, 2021*).

In this work we aim to delineate the effects of different types of soft attractive interactions on protein stability. We compare the effect of crowders that are geometrically identical but differ in their physical properties. In particular, we consider spherical "nonspecific crowders", which are capable of energetically favorable interactions with any part of the protein, and spherical hydrophobic crowders, for which these favorable interactions are limited to nonpolar (hydrophobic) amino acids. Hydrophobic crowders are expected to be destabilizing because U, but not N, will be favored by effective attractive interactions with the crowders, assuming a native conformation with a completely buried hydrophobic core. However, it is unclear *a priori* if hydrophobic crowders should be more or less destabilizing than crowders with nonspecific interactions, because the overall destabilization is determined by the net effect of crowder interactions with U and N, which will depend on the structural details of these states. For comparison, we also consider the results from excluded volume crowders, which are stabilizing due to steric repulsions. In addition to spherical crowders, we consider the crowding effect from short polypeptide chains, which can favorably interact with the protein through hydrophobic interactions and hydrogen bonding *via* their backbone NH and CO groups.

To this end, we use a coarse-grained model for folding which combines an all-atom backbone geometry with a one-bead sidechain representation, an enlarged $C_\beta$ atom. This model relies on a simplified amino acid alphabet with 3 types: polar (p), hydrophobic (h), and turn (t). Folding is driven by backbone-backbone hydrogen bonding and effective hydrophobic interactions (pairwise hh-attractions). Different sequences can be designed using simple principles (*Cordes, Davidson & Sauer, 1996*) leading the chain to adopt various protein-like folds (*Bhattacherjee & Wallin, 2012*; *Trotter & Wallin, 2020*). We focus here on a 35-amino acid sequence that folds into a stable $\alpha$-helical hairpin fold at low temperatures (*Holzgräfe & Wallin, 2015*), and is stabilized in the presence of excluded volume crowders (*Bazmi & Wallin, 2022*). We implement nonspecific interactions by making contacts between crowders and any $C_\beta$ atom on the protein (p and h amino acids) and limit the interactions to h amino acids in the case of hydrophobic crowders. For a given strength of the attraction, we find that the hydrophobic crowders are more destabilizing than the crowders interacting nonspecifically with the protein. Moreover, because the hydrophobic crowders have fewer interaction sites on the protein compared to the nonspecific crowders, the overall crowder-protein association is much weaker

for the hydrophobic crowders despite their stronger destabilizing effect. Crowders that drive protein destabilization through nonspecific interactions rely on a difference in accessible interaction sites in U and N, and therefore require a rather strong crowder-protein association at the point where they can overcome the stabilizing effect of volume exclusion.

## METHODS

### Coarse-grained model for protein folding

To model protein folding, we use the coarse-grained "$C_\beta$-model" described in *Bhattacherjee & Wallin (2012)*. It is a model with three different types of amino acids: polar (p), hydrophobic (h), and turn (t) amino acids. Geometrically, the protein chain is described using an atomistic backbone ($C_\alpha$, $C'$, N, H, $H_{\alpha 1}$, and O) and simplified sidechains using an enlarged $C_\beta$ atom. The t amino acids differ from p and h in that it does not contain a $C_\beta$ atom, which is instead replaced by an $H_{\alpha 2}$ atom. Hence, t is strongly related to glycine and more flexible than p and h. Chain conformations are completely specified by the $2N$ Ramachandran angles $\{\phi_i, \psi_i\}_{i=1}^N$. Hence, bond lengths, bond angles, and dihedral angles (*e.g.*, the peptide plane angle $\omega = 180°$) are held fixed at standard values. The potential energy function of the model, $E_p$, includes four terms $E_p = E_{loc} + E_{exvol} + E_{hbond} + E_{hp}$, described in detail in *Bhattacherjee & Wallin (2012)*. Briefly, the first term, $E_{exvol}$, provides atom-atom repulsions with a range $\sigma_a$ determined by the sum of the atoms' van der Waals radii, $\sigma_a = \sigma_i^{vdW} + \sigma_j^{vdW}$. The second term, $E_{loc}$, represents electrostatic interactions between partial charges of adjacent peptide planes. This term is included to ensure sampled $\phi_i, \psi_j$ angles agree with statistics from real protein structures (Ramachandran plots).

The two last terms, $E_{hbond}$ and $E_{hp}$, represent hydrogen bonding and effective hydrophobic interactions, respectively. These terms are essential for stabilizing and driving the formation of the native state. Hydrogen bonds are modeled using:

$$E_{hbond} = k_{hb} \sum_{ij} \gamma_{ij} \left[ 5 \left( \frac{\sigma_{hb}}{r_{ij}} \right)^{12} - 6 \left( \frac{\sigma_{hb}}{r_{ij}} \right)^{10} \right] \times (\cos\alpha_{ij} \cos\beta_{ij}) \frac{1}{2}, \tag{1}$$

where the sum is over pairs of CO, NH groups, $r_{ij}$ is the OH distance, $\sigma_{hb} = 2.0$ Å, and $k_{hbond} = 3.1$. The interaction strength is modified by a sequence-dependent scale factor $\gamma_{ij}$ taken to be 1 for hh, hp, and pp pairs, and 0.75 for tt, th, and tp pairs. The reduced hydrogen bonding capacity of t amino acids mimics the secondary structure breaking ability of glycine. The directional dependence is implemented *via* the factor $(\cos\alpha_{ij} \cos\beta_{ij})^{\frac{1}{2}}$, where $\alpha_{ij}$ and $\beta_{ij}$ are the NHO and HOC$'$ angles, respectively. Additionally, if either $\alpha_{ij} < 90°$ or $\beta_{ij} < 90°$, the ij contribution is set to zero. The hydrophobic effect is modeled using pairwise additive interactions between hydrophobic amino acids. Specifically,

$$E_{hp} = -k_{hp} \sum_{ij} g(r_{ij}; \sigma_{hp}), \tag{2}$$

where the sum is over pairs of $C_\beta$ atoms of h amino acids, excluding nearest and next-nearest neighbors along the chain, and

$$g(r; r^0) = \exp\left(-\frac{(r - r^0)^2}{2}\right) \tag{3}$$

is a function with a maximum at $r = r^0$. We set $\sigma_{hp} = 5$ Å and the interaction strength $k_{hp} = 0.805$. In this model, tuning the relative strength of hydrogen bonding and hydrophobic forces is essential to obtain folding into stable and protein-like structures (*Irbäck, Sjunnesson & Wallin, 2000*; *Irbäck, Sjunnesson & Wallin, 2001*).

## Crowders

Repulsive interactions are modeled using the pair potential (*Mittal & Best, 2010*)

$$V(r) = \left(\frac{\sigma}{r - \rho + \sigma}\right)^{12}, \tag{4}$$

where $r$ is a crowder-crowder or crowder-atom distance. Because $V(\rho) = 1$ and $V \to \infty$ as $r \to \rho - \sigma$ (for $r < \rho - \sigma$, we set $V = \infty$) the two parameters $\rho$ and $\sigma$ determine the range and "sharpness" of the repulsion, respectively. We set $\rho = 2R_c$ for crowder-crowder interactions and $\rho = \rho_{cp} = R_c + \sigma_a$ for crowder-atom interactions, where $R_c$ and $\sigma_a$ are the radii of the crowder and atom, respectively. We set $\sigma = 6$ Å for crowder-crowder interactions and $3$ Å $+ \sigma_a$ for crowder-atom interactions. A crowder-atom attractive interaction is modeled with the potential $V(r) - \epsilon_{att} g(r; \rho_{cp})$, where the function $g$ is given by Eq. (3). The form of this potential for different attraction strengths $\epsilon_{att}$ is shown in Fig. 1. For a system consisting of a protein with $N_a$ atoms and $N_{cr}$ crowder particles, the total potential energy then becomes $E = E_p + E_{cc} + E_{cp}$, where

$$E_{cc} = \sum_{i}^{N_{cr}} \sum_{j=i+1}^{N_{cr}} V(r_{ij}) \tag{5}$$

and

$$E_{cp} = \sum_{i}^{N_{cr}} \left[ \sum_{j}^{N_a} V(r_{ij}) - \epsilon_{att} \sum_{j} g(r_{ij}; \rho_{cp}) \right] \tag{6}$$

are the crowder-crowder and crowder-protein energies, respectively. In Eq. (6), the second sum within square brackets controls which atoms are subject to crowder attraction. For crowders with nonspecific attraction to the protein, the sum is over all $C_\beta$ atoms (p and h amino acids). For hydrophobic crowder, the sum is over the $C_\beta$ atoms of h amino acids. For excluded volume crowders, $\epsilon_{att} = 0$.

## Simulated tempering Monte Carlo

To find the thermodynamic behavior of various protein-crowder systems, as determined by the amino acid sequence, number of crowders, and the energy function $E(r)$, we use simulated tempering Monte Carlo (MC) (*Marinari & Parisi, 1992*; *Lyubartsev et al., 1992*; *Irbäck & Potthast, 1995*). In addition to a random walk in conformational space, as in basic
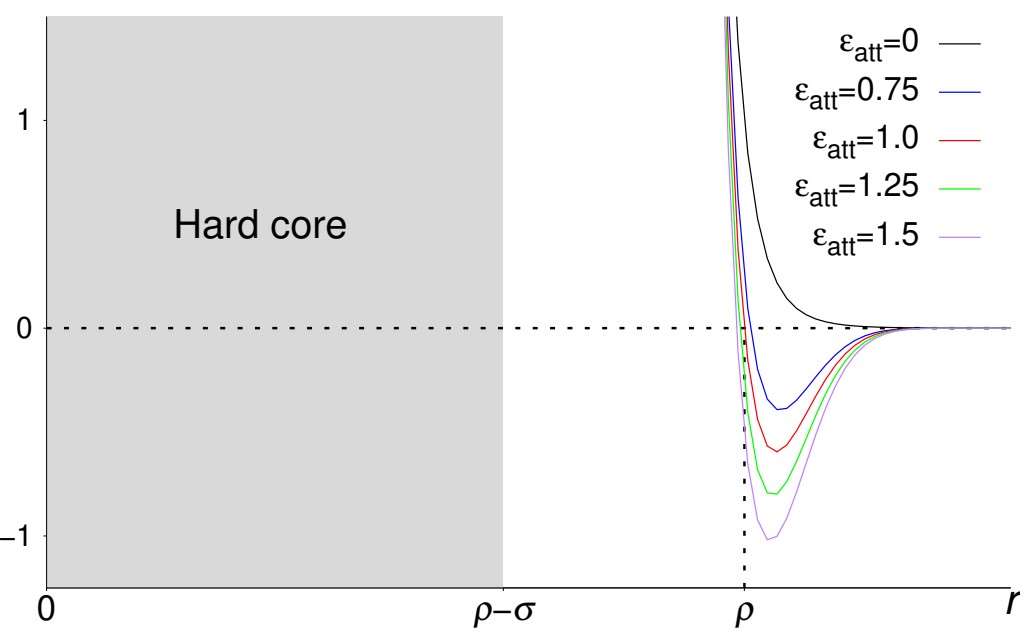

**Figure 1** **Pair potentials used to describe crowder-crowder and crowder-atom interactions in this work.** Shown is $V(r) - \epsilon_{att}g(r)$ for different strengths $\epsilon_{att}$ of the attractive part of the potential, $-\epsilon_{att}g(r)$. The repulsive part, $V(r)$, is controlled by the two parameters $\rho$ and $\sigma$. For distances $r < \rho - \sigma$, $V = \infty$, meaning that all pair potentials include a hard core region (gray shaded area). The functional forms of $g(r)$ and $V(r)$ are given in Eqs. (3) and (4), respectively.

Monte Carlo, simulated tempering also carries out a random walk in temperature while keeping the simulation at equilibrium. This is achieved by defining a set of temperatures, $\{T_j\}_{j=1}^M$, and simulating the joint probability distribution

$$P(r,j) \propto e^{-\beta_j E(r) + g_j},\tag{7}$$

where $\beta_j = 1/k_B T_j$, $k_B$ is Boltzmann's constant, and $j$ has been made a dynamic parameter. The $g_j$'s are M simulation parameters that control the marginal distribution $P(j)$. Jumps between temperatures, $j \rightarrow j'$, are accomplished as MC updates, with acceptance probability

$$P_{acc}(r, j \rightarrow j') = \min\left[1, e^{-E(r)\left(\beta_{j'} - \beta_j\right) + g_{j'} - g_j}\right].\tag{8}$$

A common choice for the $g_j$ parameters, which we follow here, is to select them such that $P(j)$ is roughly flat, ensuring that sampling of conformations takes place at each temperature $T_j$.

## Simulations and analysis details

Equilibrium behaviors of crowder-peptide systems are determined using simulated tempering Monte Carlo simulations (*Marinari & Parisi, 1992*). Runs are carried out with either 8 temperatures in the range $k_B T = 0.48$–$0.68$ or 10 temperatures in the range $0.40$–$0.70$. For each system, 10 independent runs with $5 \times 10^9$ elementary MC updates are
performed. Initial configurations are obtained by picking random chain conformations and random spherical crowder positions, followed by a relaxation step to remove any hard-core steric clashes. Monte Carlo chain updates are divided equally between the protein chain and the crowder particles. Two different types of chain moves are used: biased Gaussian steps (BGS) (*Favrin, Irbäck & Sjunnesson, 2001*), which produce approximately local chain deformations, and pivot moves, which produce global changes. In the latter type, a single $\phi_i$ or $\psi_i$ angle is set to a new random value such that the chain rotates around the $NC_\alpha$ or $C_\alpha C'$ bonds. The frequencies of chain updates are set so that BGS is most frequent at low $T$s and pivot is most frequent at high $T$s. Temperature updates are attempted every 100 MC step. Spherical crowder simulations are carried out in a cubic box of side length $L = 100$ Å while for simulations of polypeptide crowders $L = 65$ Å. The number of spherical crowding agents are 14, 28, 42, and 56, corresponding to packing fractions $\phi_c = 0.10$, 0.20, 0.30, and 0,40, respectively. The crowder radius is $R_c = 12$ Å. For simulation carried out with 8 temperatures in the range $k_B T = 0.48$–$0.68$, the multistate Bennett acceptance ratio (MBAR) technique was applied to determine thermodynamic averages in the range $k_B T = 0.40$–$0.70$.

## Observables

We quantify native state stability using two quantities, $\Delta F$ and $T_m$. The free energy of folding is determined using

$$\Delta F = F_N - F_U = -k_B T \ln \frac{P_{nat}}{1 - P_{nat}}, \tag{9}$$

where $F_U$ and $F_N$ are the free energies of the unfolded and native states, respectively, and $P_{nat}$ is the population of the native state. The native state population $P_{nat}$ is determined as in Ref. *Bazmi & Wallin (2022)*. Conformations are considered part of the native state if $Q_{nat} \geq Q_{cut}$ where $Q_{nat}$ is the number of formed native contacts and $Q_{cut} = 50$. The folding midpoint temperature, $T_m$, is determined by fitting the temperature dependence of $P_{nat}$ to a two-state model with two fit parameters.

## RESULTS

### Spherical crowders

Using MC simulations, we determined the thermodynamics behavior of our model protein, $\alpha_{35}$, in isolation and in the presence of three different types of spherical crowders having the same radius ($R_c = 12$ Å) but different interactions with the protein chain: (1) excluded volume crowders, capable of only steric repulsions; (2) "nonspecific" crowders, which in addition to steric repulsions, form energetically favorable contacts with polar and nonpolar amino acids; and (3) hydrophobic crowders, which differ from the nonspecific crowders in that they interact favorably with nonpolar amino acids only. We varied the crowder volume fraction, $\phi_c$, and the strength $\epsilon_{att}$ of the attractive protein-crowder interactions. In general, we find that the equilibrium stability of the $\alpha_{35}$ native state, as quantified by the free energy of folding, $\Delta F = -k_B T \ln[P_{nat}/(1 - P_{nat})]$, where $P_{nat}$ is the population of the native state, depends on both $\phi_c$ and $\epsilon_{att}$. As an illustration of our results, Fig. 2 shows the temperature dependence of $\Delta F$ at $\epsilon_{att} = 1.5$, for different $\phi_c$ and crowder types.

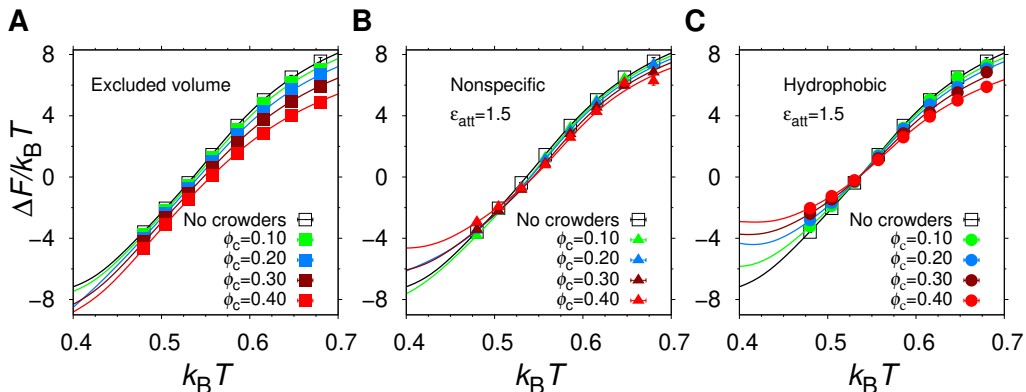

**Figure 2** **The free energy of folding.** Temperature dependence of the free energy of folding of $\alpha_{35}$ in the absence and presence of (A) excluded volume crowders, (B) nonspecific crowders, and (C) hydrophobic crowders, at different volume fractions, $\phi_c$, and (B, C) for one strength of the crowder-protein attractions, $\epsilon_{att} = 1.5$.

In the absence of crowders (solid black curve in Fig. 2), the helical hairpin fold of $\alpha_{35}$ is highly stable at low $T$. For example, at the lowest studied temperature ($k_B T = 0.40$), which corresponds to $T \approx 0.75 T_m^0$, where $T_m^0$ is the midpoint temperature of the $\alpha_{35}$ folding transition for $\phi_c = 0$, $\Delta F/k_B T = -7.1$, which is within the range of stabilities typically found for single-domain proteins (*Zeldovich, Chen & Shakhnovich, 2007*). The midpoint temperature $T_m^0$, *i.e.,* the temperature for which $\Delta F = 0$, is also commonly used as a measure of protein stability. For $\alpha_{35}$, $k_B T_m^0 = 0.535$, as obtained previously (*Bazmi & Wallin, 2022*) by fitting the of $\alpha_{35}$ folding curve *i.e.,* $P_{nat}$ as function of $T$, to a two-state model with two free parameters. For the values of $\epsilon_{att}$ considered here, $\alpha_{35}$ remains stable in the low $T$ region even when the crowder packing fraction reaches $\phi_c = 0.40$. However, there are clear variations between different crowder types as can be seen in Fig. 2. For example, for $\phi_c = 0.30$ and $k_B T = 0.40$, the excluded volume crowders give $\Delta F/k_B T = -8.3$, which is a decrease relative to the $\phi_c = 0$ case. For the nonspecific and hydrophobic crowders the corresponding $\Delta F/k_B T$ values are $-6.1$ and $-3.3$, respectively, which, by contrast, are increases relative to $\phi_c = 0$.

At a temperature just below the folding midpoint, $T^- = 0.95 T_m^0$, we observe the following trends. Upon the addition of excluded volume crowders, $\Delta F$ decreases monotonically with $\phi_c$, as shown in Figs. 3A and 3C, meaning a strict stabilization of the protein. The soft interactions of the nonspecific and hydrophobic crowders are expected to be destabilizing. However, because these two crowder types occupy the same space as the excluded volume crowders, they also provide a stabilizing effect. The net effect will be determined by a competition between the stabilizing steric repulsions and the destabilizing soft interactions. Indeed, at low attraction strength, $\epsilon_{att} = 1.0$, the addition of either nonspecific or hydrophobic crowders leads to an overall stabilization of $\alpha_{35}$ (see Fig. 3A and 3C). At $\epsilon_{att} = 1.5$, the hydrophobic crowders overcome the stabilizing excluded volume effect leading to a net destabilization, while the nonspecific crowders are still net stabilizing (see Figs. 3B and 3D), although for the nonspecific crowders $\Delta F \approx 0$ at $\phi_c = 0.40$. The

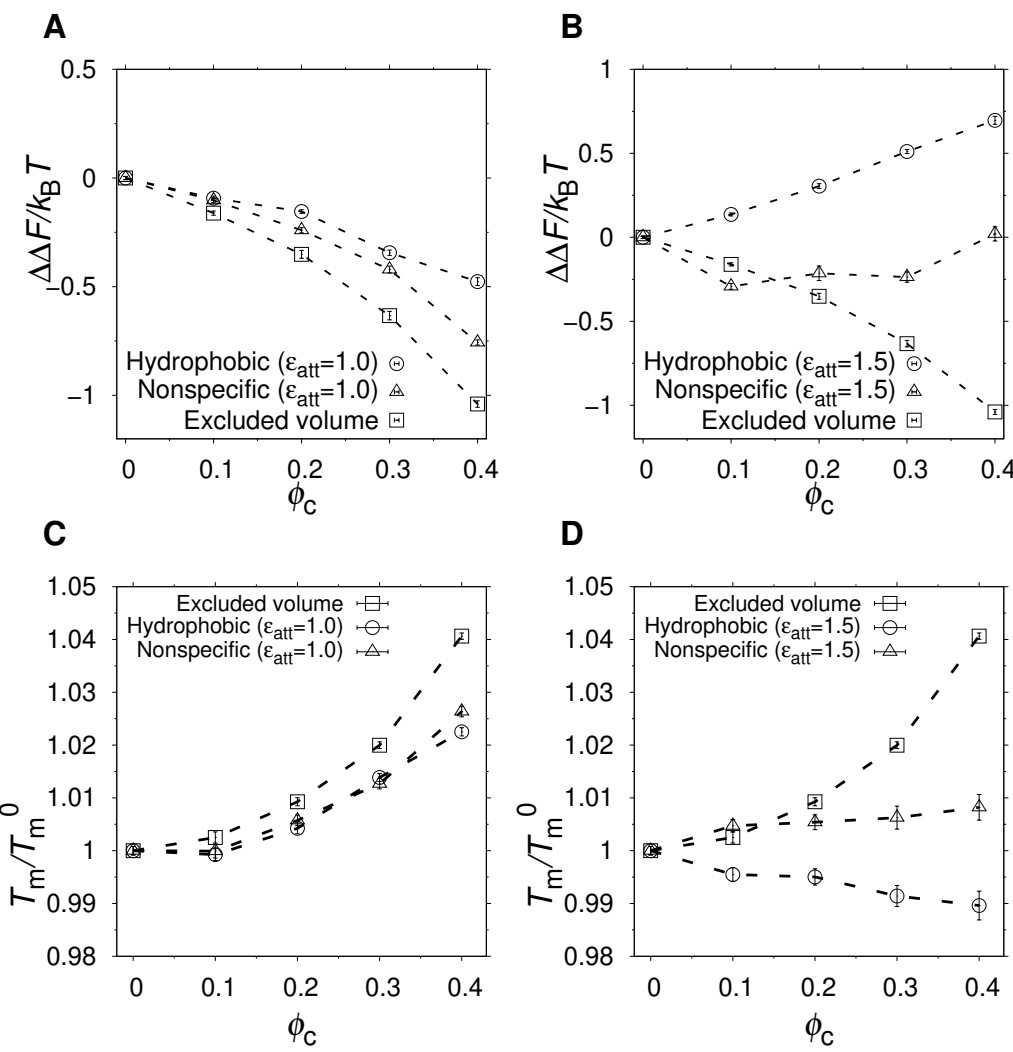

**Figure 3 Effect of crowding on the native state stability.** The change in free energy of folding, $\Delta\Delta F(\phi_c) = \Delta F(\phi_c) - \Delta F_0$, where $\Delta F_0$ is the folding free energy in the absence of crowders, as a function of $\phi_c$, for excluded volume (squares), nonspecific (triangles), and hydrophobic (circles) crowders with attraction strengths (A) $\epsilon_{att} = 1.0$ and (B) $\epsilon_{att} = 1.5$. All free energies are obtained at $T^- = 0.95T_m^0$, where $T_m^0$ is the midpoint temperature at $\phi_c = 0$ ($k_B T^- = 0.508$). Relative change in midpoint folding temperature, $T_m/T_m^0$, as function of $\phi_c$, for (C) $\epsilon_{att} = 1.0$ and (D) $\epsilon_{att} = 1.5$. Dashed lines between points are drawn to guide the eye.

picture obtained is similar if instead $T_m$ is used to quantify native state stability. As shown in Fig. 3C, there is a monotonic increase in $T_m$ for both the excluded volume crowders and for the hydrophobic and nonspecific crowders with $\epsilon_{att} = 1.0$. For $\epsilon_{att} = 1.5$, the nonspecific crowders still exhibit a monotonic increase in $T_m$ with increases $\phi_c$ while the hydrophobic crowders instead exhibit a monotonic decrease (see Fig. 3D). Overall, our results show that excluded volume crowders provide a stabilizing effect on $\alpha_{35}$, which can be counteracted by the presence of soft attractive interactions. The hydrophobic crowders provide a stronger destabilizing effect than the nonspecific crowders.

## How can the stronger destabilizing effect of hydrophobic attractions be explained?

To address the question of why hydrophobic attractive interactions are more destabilizing than nonspecific interactions, we consider the interaction energy between crowders and protein, $E_{cp}$ (see Methods). $E_{cp}$ is a mix of repulsive and attractive energy contributions. We consider, in particular, $E_{cp}^U$ and $E_{cp}^N$, *i.e.,* the crowder-protein energy determined separately for the U and N states. When the difference $\Delta E_{cp} = E_{cp}^U - E_{cp}^N < 0$ there is a net energetic stabilization of U, which should have a destabilizing effect on the protein.

Figures 4A and 4B show the $\phi_c$-dependence of $E_{cp}^U$ and $E_{cp}^N$ for two different values of $\epsilon_{att}$. For hydrophobic crowders and $\epsilon_{att} = 1.0$, $E_{cp}^N$ (open circles) exhibits a slight positive curve and $E_{cp}^U$ a slight negative curve. These trends are in line with an expanded U with some nonpolar amino acids available for favorable interactions with the crowders, and an N with a hydrophobic core well shielded from the crowders. For $\epsilon_{att} = 1.5$, the trend is similar but $E_{cp}^N$ now exhibits a slight negative curve, indicating that, at this strength of attractions, N becomes slightly distorted to accommodate favorable contacts with the hydrophobic crowders. However, these crowder-proteins interactions are rather limited. At the highest packing fraction, $\phi_c = 0.40$, $E_{cp}^N \approx -1.5$, corresponding to $\approx$ 1–2 fully formed contacts between an nonpolar amino acid and a crowder. In comparison, for the nonspecific case, interactions between crowders and protein are much more prevalent for both U and N. $E_{cp}^U$ is a sharply decreasing function of $\phi_c$ for both $\epsilon_{att} = 1.0$ and 1.5, which will strongly stabilize U at high $\phi_c$. However, the decrease in $E_{cp}^U$ is nearly fully compensated by a decrease in $E_{cp}^N$. Comparing the two types of crowders, we find that the net energetic effect, $\Delta E_{cp} = E_{cp}^U - E_{cp}^N$, which is what drives destabilization, is more negative for the hydrophobic crowders than for the nonspecific crowders, as can be seen in Figs. 4C and 4D.

Hence, an interesting difference between the two crowder types is that, while their net effects on stability are similar, they differ substantially in the degree of association with the protein. While hydrophobic crowders associate weakly with the protein, even in U, the protein-crowder association in the case of nonspecific interactions is greater by approximately an order of magnitude, as quantified by the magnitude of the interaction energies in the native state, $E_{cp}^N$ (see Figs. 4A and 4B).

## Balancing destabilizing soft interactions and stabilizing steric repulsions

The $\alpha_{35}$ protein is net stabilized by the presence of hydrophobic crowders at $\epsilon_{att} = 1.0$ but net destabilized at $\epsilon_{att} = 1.5$. This suggests a critical attraction strength at which there is no net change in stability. Indeed, for $\epsilon_{att} = 1.25$, the relative change in $T_m$ is at most one percent and the change in folding free energy, $\Delta\Delta F$, is very close to zero, as shown in Figs. 5A and 5B. Interestingly, this balance between stabilizing and destabilizing effects is largely independent of $\phi_c$. The crowding effect on stability at any given $\epsilon_{att}$ is, however, generally dependent on $T$, as shown in Fig. 5C for $\Delta\Delta F$. There is a $T$-dependence also for excluded volume crowders, which are strictly stabilizing ($\Delta\Delta F < 0$). Such a $T$-dependence is possible for excluded volume crowders despite their lack of energetically favorable energetic interactions with the protein, because the size of U is not constant but vary with

Peer⌡

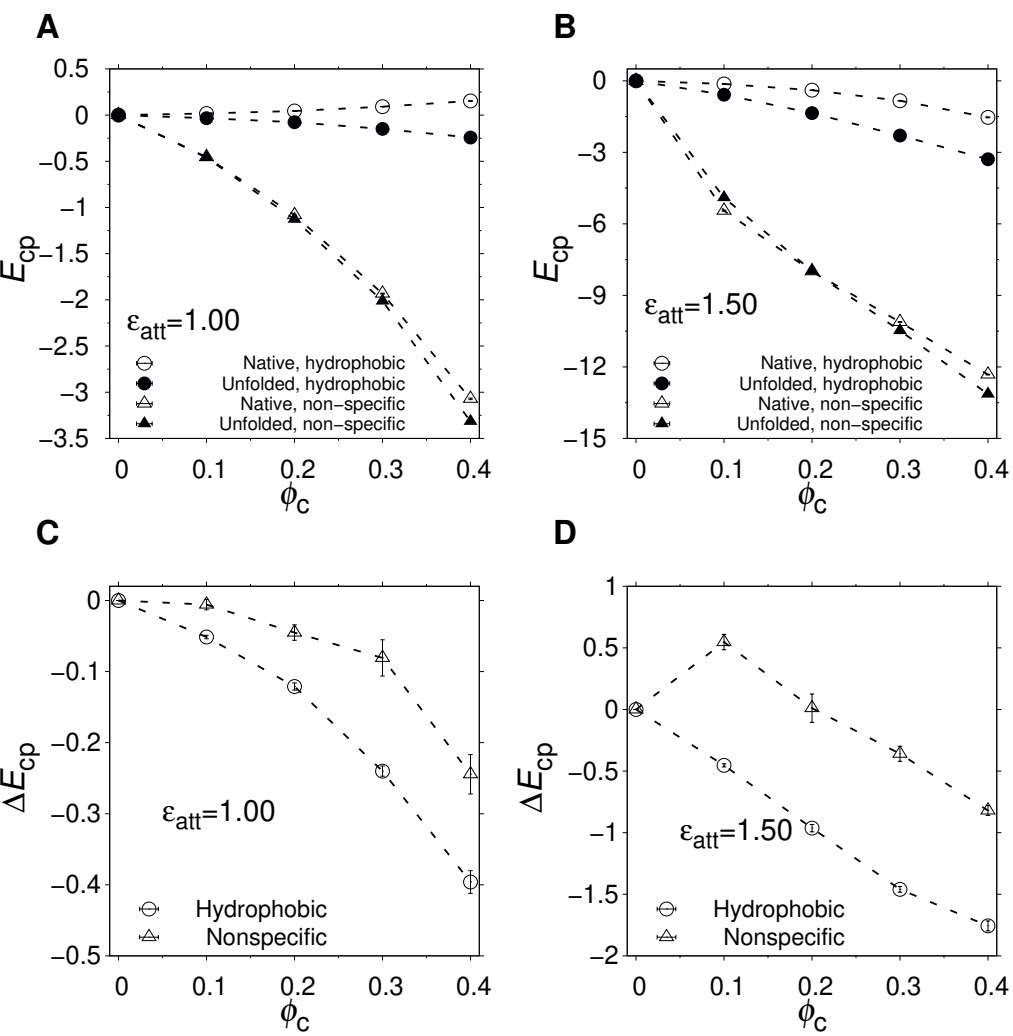

**Figure 4** **Crowder-protein interaction energies.** (A and B) $\phi_c$-dependence of the average crowder-protein interaction energy, $E_{cp}$, determined separately for the native state ($E_{cp}^N$; open symbols) and the unfolded state ($E_{cp}^U$; solid symbols) states. Results are shown for nonspecific (triangles) and hydrophobic (circles) crowders at two different crowder-protein interaction strengths, $\epsilon_{att}$. (C and D) $\phi_c$-dependence of the change in the interaction energy, $\Delta E_{cp} = E_{cp}^U - E_{cp}^N$. The temperature is the same as in Fig. 3. Dashed lines between points are drawn to guide the eye.

*T* (*Bazmi & Wallin, 2022*). For hydrophobic crowders, due to the energetic stabilization of U relative to N, at low enough temperatures, the net effect crosses over from stabilizing to destabilization, *i.e.,* $\Delta\Delta F$ changes from negative to positive. This crossover temperature (*Zhou, 2013*), $T_c$, increases with the contact strength $\epsilon_{att}$, as shown in Fig. 5D. Our findings for $\alpha_{35}$ are similar to previous studies on other proteins, such as ubiquitin, which showed a crossover temperature in the presence of either synthetic polymers or protein crowders (*Wang et al., 2012b*; *Zhou, 2013*).

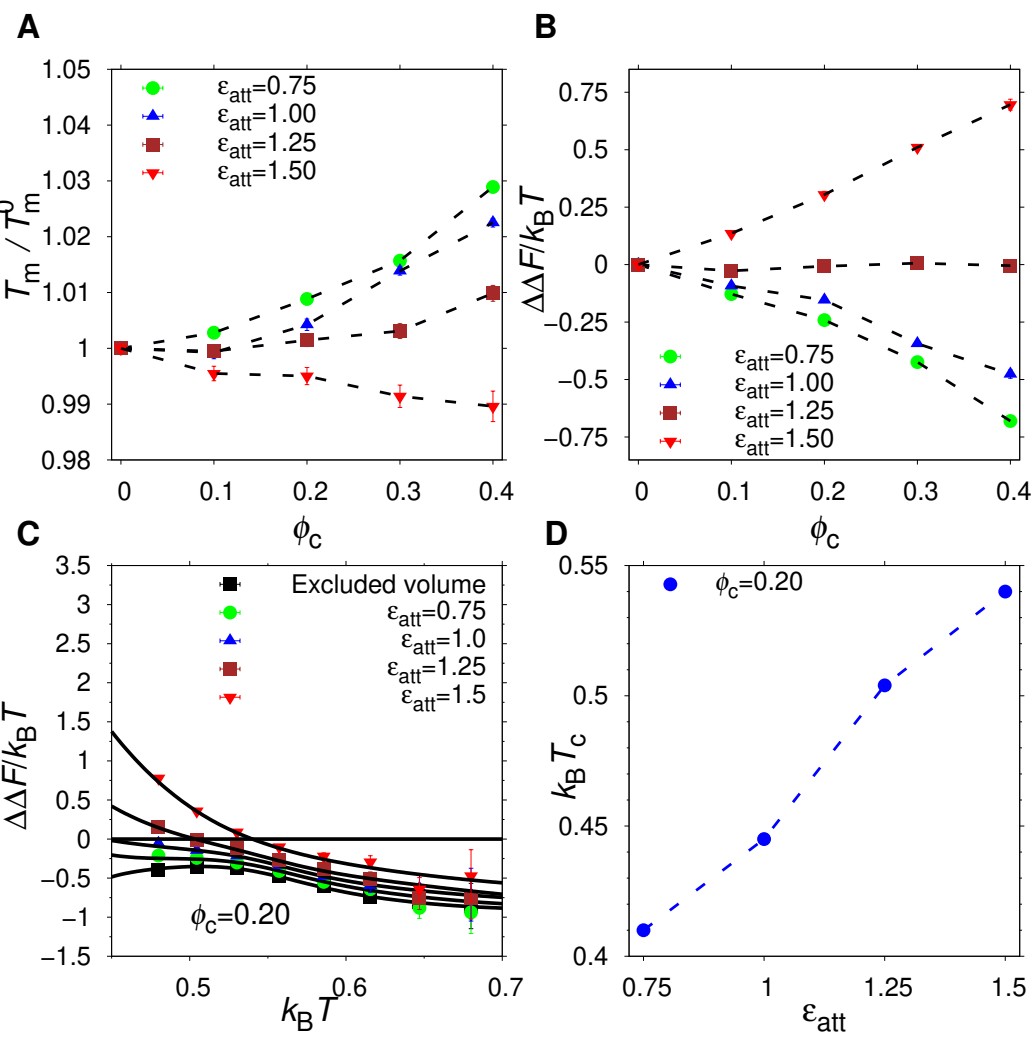

**Figure 5  Effect of hydrophobic crowders on the native state stability.** (A) Change in the folding mid-point temperature, and (B) change in the folding free energy at $T^-$, as functions of $\phi_c$ for different attraction strengths $\epsilon_{att}$. (C) $\Delta\Delta F$ as a function of temperature for difference values of $\epsilon_{att}$. A crossover temperature, $T_{cross}$ is defined by $\Delta\Delta F = 0$. (D) $T_{cross}$ as a function of attraction strengths for fixed volume fraction $\phi_c = 0.20$. Dashed lines between points are drawn to guide the eye.

## Polypeptide crowders

We turn now to the folding of $\alpha_{35}$ in the presence of polypeptide chains. We use the same model for the peptides as for $\alpha_{35}$, with the restriction that energetically favorable interactions (*i.e.,* hydrophobic and hydrogen bond interactions) between crowding peptides are turned off. This avoids the formation of oligomeric peptide structures, which would complicate the analysis. We consider two different 5-amino acid sequences: ppppp (peptide 1) and pphpp (peptide 2). Both peptides can thus interact with the $\alpha_{35}$ chain through backbone-backbone hydrogen bonding and peptide 2, due to its central h amino acid, can additionally interact with $\alpha_{35}$ through hydrophobic attractions.

Figures 6A and 6B show the $T$-dependence of the free energy of folding, $\Delta F$, for different numbers $N_{\text{pep}}$ of peptide 1 or peptide 2 chains added to the system. For $N_{\text{pep}} \leq 42$, peptide 1 induces a weak stabilization at high $T$ and weak destabilization at low $T$. This behavior is consistent with a competition between entropy-driven stabilization due to volume exclusion by the peptides and energy-driven destabilization due to inter-chain hydrogen bonding. Interestingly, there is a rather broad temperature range around $T_{\text{m}}^0$ ($k_{\text{B}}T \approx$ 0.45–0.60) with no detectable change in $\Delta F$ (see Figs. 6A and 6D). The apparent lack of crowding effects in this temperature range can arise either because (i) the crowding effects are overall weak at the studied peptide concentrations or (ii) the two opposing crowding effects, peptide excluded volume and peptide-protein attractions, are equal in magnitude and therefore cancel out. To determine which scenario holds, we performed additional simulations with peptides that were only allowed to interact with other chains *via* repulsive interactions. These "excluded volume peptides" significantly stabilize $\alpha_{35}$ across all $T$s (see Figs. 6C and 6D), which means that scenario (ii) above holds. Hence, in a relatively broad range around the midpoint temperature of the protein, the excluded volume effect due to the peptides is almost perfectly counteracted by soft interactions in the form of hydrogen bonds.

Peptide 2 provides a stronger destabilizing effect on $\alpha_{35}$ than peptide 1 at a given concentration, as seen in Figs. 6B and 6D. We therefore study up to $N_{\text{pep}} = 28$ peptide 2 chains. At $T \approx T^-$, $\Delta F$ monotonically increases with the number of added peptide 2 chains in contrast to the flat $\Delta F$ exhibited by peptide 1. At very low $T$ and $N_{\text{pep}} = 28$, peptide 2 even leads $\alpha_{35}$ to become net unstable ($\Delta F > 0$), although the error bars are larger at the lowest studied $T$s. A similar behavior is seen for peptide 1 and $N_{\text{pep}} = 56$. Structurally, low-energy conformations obtained for the peptide 1 and peptide 2 systems exhibit non-native features, including $\beta$-sheet structure. An example of a low-energy $\alpha_{35}$ structure in the precence of peptide 2 crowders is given in Fig. 7A. The destabilization at low $T$ is, at least in part, driven by energetically favorable crowder-peptide interactions that also causes non-native structures. By contrast, low-energy conformations remain native-like in the case of excluded volume peptides (see Fig. 7B). It is instructive to get a rough idea of the volume fraction $\phi_{\text{c}}$ occupied by the peptides in our systems. Assuming each atom in our $C_\beta$-model occupies volume according to its van der Waals radius, we obtain $\phi_{\text{c}} \approx 0.07$ for $N_{\text{pep}} = 28$. Alternatively, if we assume amino acids are spheres with radius 3.8 Å (typical $C_\alpha$-$C_\alpha$ distances of peptide bonds) we obtain a slightly higher value, $\phi_{\text{c}} \approx 0.12$. What makes peptide 2 more destabilizing than peptide 1? Because the two peptides are geometrically identical, any difference must derive from soft interactions. Figures 8A and 8B show the peptide concentration dependence of the quantity $\Delta E_{\text{cp}} = E_{\text{cp}}^{\text{U}} - E_{\text{cp}}^{\text{N}}$ (Fig. 4), determined separately for peptide (crowder)-protein hydrogen bond ($\Delta E_{\text{cp}}^{\text{hbond}}$) and hydrophobic ($\Delta E_{\text{cp}}^{\text{hp}}$) energies. For peptide 1, $\Delta E_{\text{cp}}^{\text{hp}} = 0$, since this peptide lacks a hydrophobic amino acid. For peptide 2, $\Delta E_{\text{cp}}^{\text{hp}} < 0$ for all peptide-protein systems. This means that, vis-à-vis hydrophobic interactions, peptide 2 interacts more favorably with U than with N, causing $\alpha_{35}$ destabilization. The formation of hydrogen bonds between peptides and the protein are also destabilizing because generally $\Delta E_{\text{cp}}^{\text{hbond}} < 0$. Interestingly, $\Delta E_{\text{cp}}^{\text{hbond}}$ is more negative for peptide 2 than for peptide 1, such that the presence of a hydrophobic amino acid on

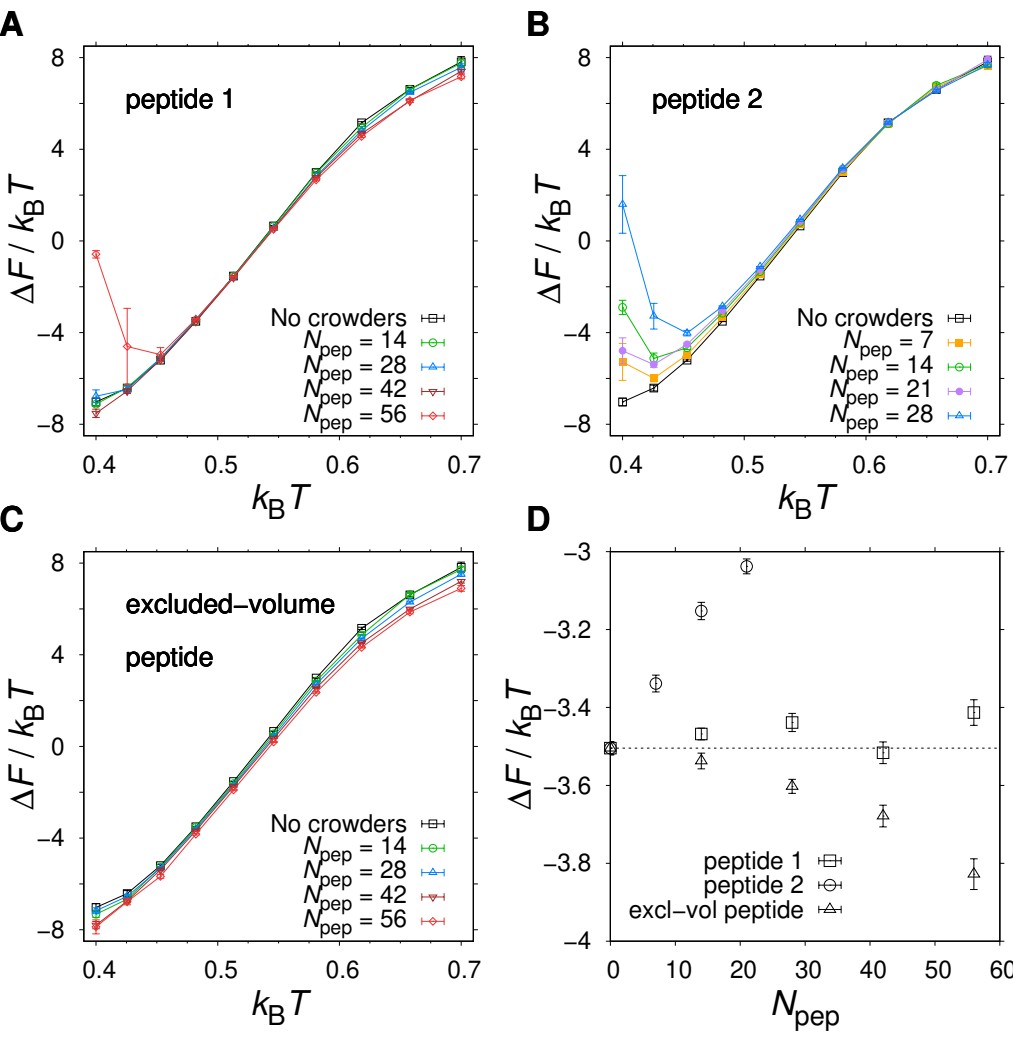

**Figure 6** **Native state stability of $\alpha_{35}$ in the presence of different types of polypeptide crowders.** Folding free energy, $\Delta F$, as function of temperature $T$ for three peptide crowder types: (A) peptide 1 with sequence ppppp, (B) peptide 2 with sequence pphpp, and (C) a peptide geometrically identical to peptides 1 and 2 but with inter-chain hydrophobic and hydrogen bond interactions turned off. (D) $\Delta F$ as function of the number of peptides, $N_{\text{pep}}$, at $T \approx T^{-}$.

peptide 2 also enhances the formation of hydrogen bonds with the protein chain. In other words, there is a cooperative effect between hydrophobic and hydrogen bond interactions in peptide 2, which further enhances the destabilizing effect of this peptide relative to peptide 1 which has only polar amino acids.

## DISCUSSION

We have used a sequence-based coarse-grained protein model to study crowding-induced changes to the stability of a model protein with a sequence that folds to an $\alpha$-helical hairpin at low temperatures. In particular, we investigated crowders making different types of soft interactions with the protein, *i.e.,* interactions different from hard-core steric repulsions.

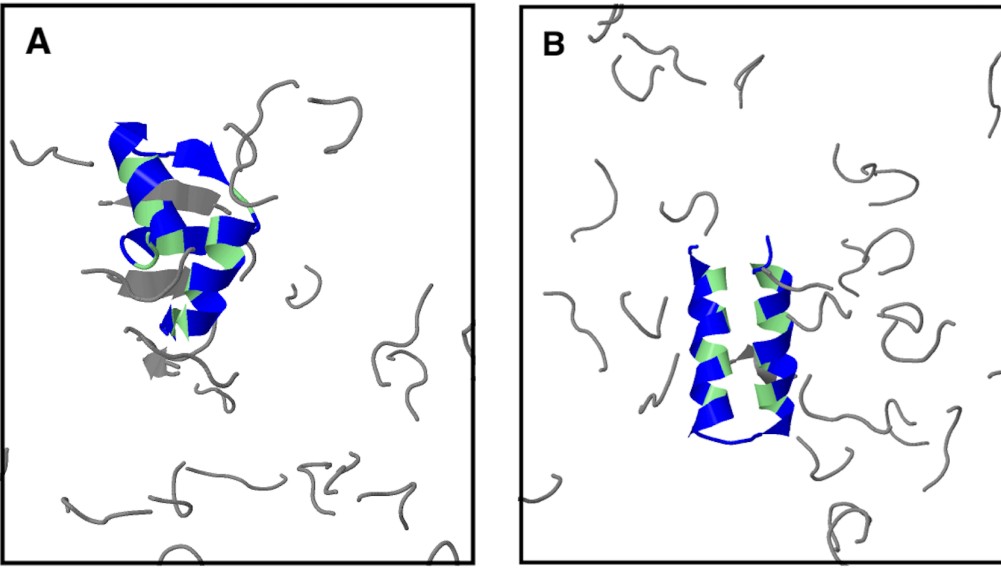

**Figure 7** **Protein $\alpha_{35}$ in the presence of two different types of peptide crowders: (A) peptide 2 and (B) excluded-volume peptides.** The structures shown are taken from simulations with 28 peptide crowders at the lowest studied temperature ($k_B T = 0.40$). Protein (hydrophobic amino acids in green, others in blue) and peptides (gray) are shown in cartoon representation. For a description of the two types of peptide crowders, see text.

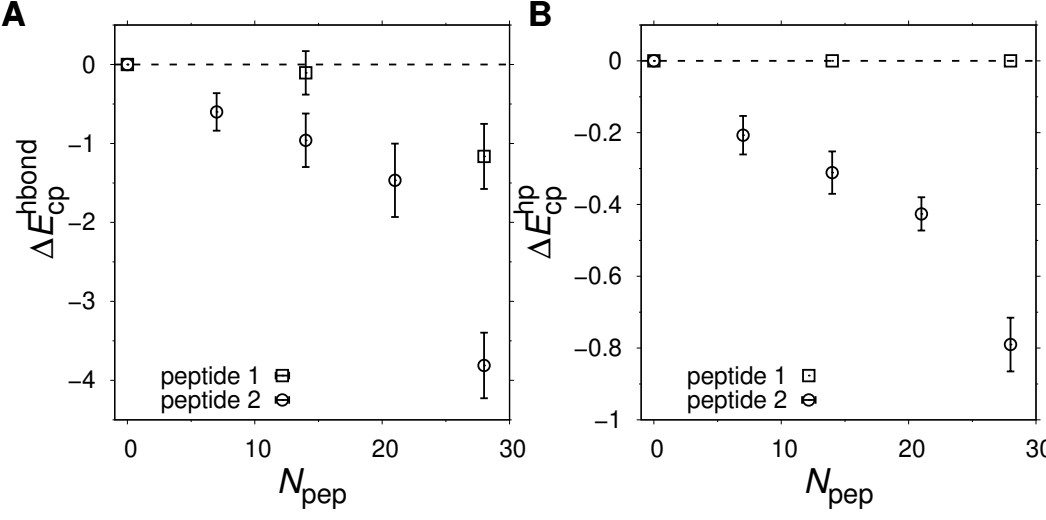

**Figure 8** **Energetically favorable interactions between crowder peptides and the protein.** Hydrophobic and hydrogen-bond peptide-protein interaction energies measured for the native ($E_{cp}^{N,hp}$ and $E_{cp}^{N,hbond}$) and unfolded ($E_{cp}^{U,hp}$ and $E_{cp}^{U,hbond}$) states. Shown are (A) $\Delta E_{cp}^{hbond} = E_{cp}^{U,hbond} - E_{cp}^{N,hbond}$ and (B) $\Delta E_{cp}^{hp} = E_{cp}^{U,hp} - E_{cp}^{N,hp}$ as function of $N_{pep}$. Negative $\Delta E_{cp}$ values means that the peptides interact more favorably with U relative to N. The temperature is the same as in Fig. 6D.

We found that crowders with a hydrophobic character, *i.e.,* crowders interacting favorably with only nonpolar amino acids, provide a stronger destabilizing effect compared to crowders that interact nonspecifically with the protein, *i.e.,* make favorable contacts with both polar and nonpolar amino acids. Both types of soft interactions are counteracting the stabilizing effect of excluded volume. For weak attraction strengths, these crowders still increase the stability of the protein relative to the dilute limit where there are no crowding effects. At a critical strength of the attraction, the stabilizing and destabilizing effects cancel leading to a net zero change in protein stability over a wide range of crowder concentrations. Similar results were obtained in a structure-based one-bead-per-amino acid model studied by Mittal et al. (*Kim, Bhattacharya & Mittal, 2014*).

The destabilizing effect of either hydrophobic or nonspecific crowder-protein interactions arises because interactions with U are generally more energetically favorable than interactions with N. This is manifested by the net crowder-protein interactions, $\Delta E_{cp}$, which is negative for high $\phi_c$ for both crowder types. However, perhaps counterintuitively, the overall association of the hydrophobic crowders with the protein is much less pronounced than for the nonspecific crowders, at a given degree of destabilization. This is illustrated schematically in Fig. 9. Quantitatively, we find that the magnitude of the interaction energy between crowders and protein, $E_{cp}$, is larger for the nonspecific crowders than for the hydrophobic crowders by roughly an order of magnitude. The reason for the difference is the way the two crowder types achieve destabilization. Nonspecific crowders rely on a difference between two rather large favorable interaction energies for U and N. Hydrophobic crowders, for which favorable interactions with N are almost absent due to hydrophobic amino acids being buried in the native structure, obtain destabilization even for relatively weak favorable interactions with U. This difference between hydrophobic and nonspecific soft interactions might be tested experimentally using protein crowders, and varying the chemical nature of surface-exposed amino acids through mutations.

Our results also confirm that the addition of crowders attracted to a protein can lead to temperature-dependent crowding effects, as demonstrated experimentally (*Wang et al., 2012b*). Specifically, we found that the our model protein exhibits a crossover temperature, $T$cross, below which the crowders are destabilizing and above which the crowders are stabilizing. The presence of such a crossover temperature is well established in the literature and has been observed in the presence of both synthetic polymer crowders and protein crowders (*Wang et al., 2012b*; *Zhou, 2013*; *Timr & Sterpone, 2021*). The origin of this crossover temperature is believed to be due to the fact that energetic effects are becoming more important at low $T$, while at high $T$ entropic effects dominate (*Zhou, 2013*; *Timr & Sterpone, 2021*). It should be noted that the hydrophobic effect is itself partly entropically driven, while in our model it is treated as an effective energetically driven interaction. Destabilization driven entirely by soft hydrophobic interactions are not guaranteed to increase with decreasing $T$.

We have also investigated how the folding and stability of our model protein are impacted by short polypeptide chains, which differ from our spherical crowder in that they are able to interact with the protein through hydrogen bonding. Results for this system indicate an interplay between hydrogen bonds and hydrophobic interactions. Peptide chains with

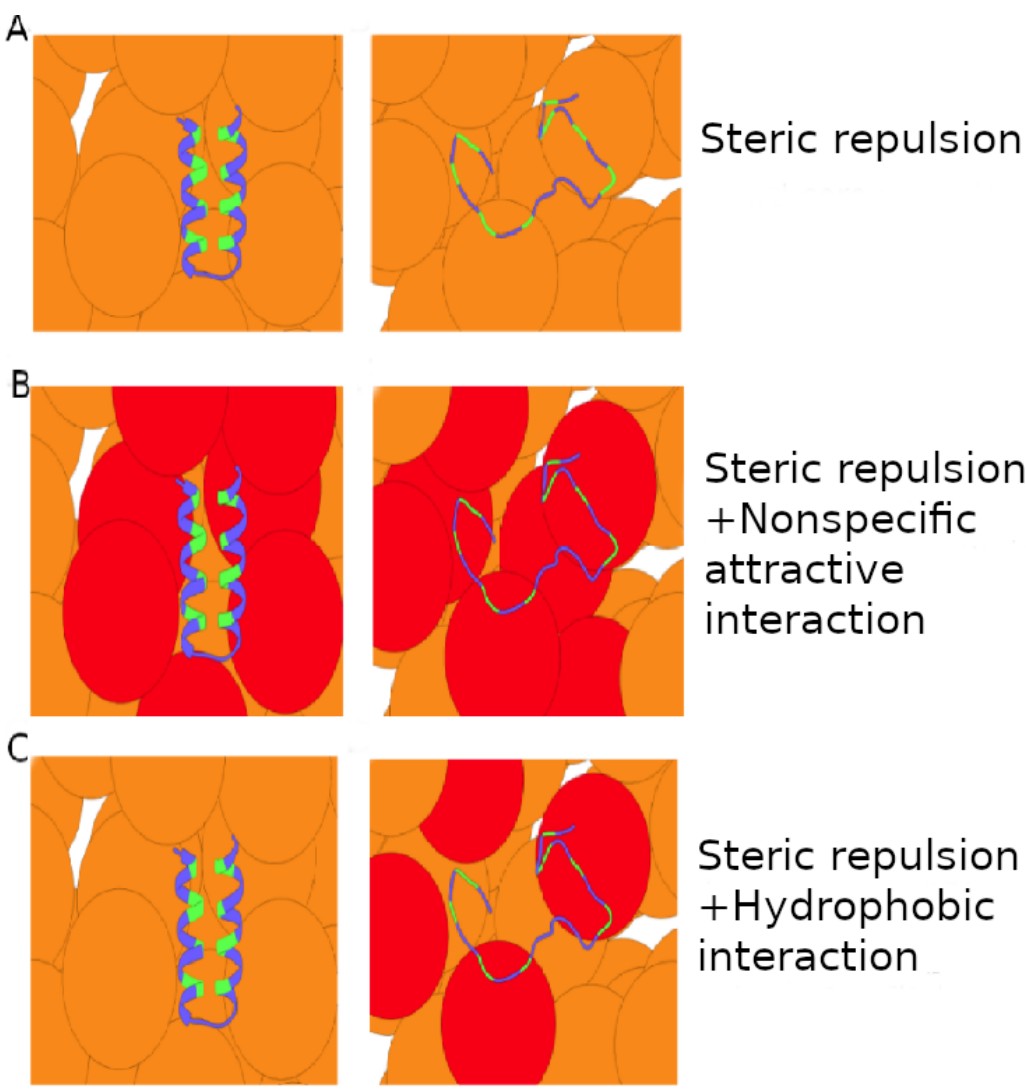

**Figure 9  Schematics of attractive crowder-protein interactions.** Crowder molecules (orange and red spheres) interacting with a two-state protein (ribbon) through (A) hard-core steric repulsions, (B) nonspecific attractive interactions, and (C) hydrophobic interactions. Crowders (red) make energetically favorable contacts with unfolded and native protein conformations in the case of nonspecific interactions, and with only unfolded conformations in the case of hydrophobic interactions if the native conformation has a core of hydrophobic amino acids (green) inaccessible to other molecules.

partial hydrophobic character exhibit much stronger destabilizing effects than peptides with only polar amino acids. Part of this destabilization is due to hydrophobic interactions directly stabilizing U, in the same way as found for the hydrophobic spherical crowders. However, we found that the presence of a hydrophobic amino acid in the sequence of our peptide crowder also tends to promote the formation of additional hydrogen bonds with the protein (see Fig. 8), thus further strengthening the destabilization. It should be pointed out, however, that our polypeptide crowders are relatively short. It would be interesting to test the crowding effect of much longer polypeptide chains in our model, which could

mimic macromolecular crowders capable of both substantial excluded volume effects and soft interactions in the form of hydrophobic interactions and hydrogen bonding.

## CONCLUSIONS

In summary, we have used a sequenced-based model to study the effects of attractive interactions between crowders and protein on the stability of the protein's native state. Our results highlight the importance of considering both the type and the strength of soft crowder-protein interactions when evaluating the impact of crowding. Our main conclusions are as follows: (1) soft attractive interactions have a generally destabilizing effect on protein stability in the context of both spherical and polypeptide chain crowders; (2) hydrophobic protein-crowder interactions provide a stronger destabilizing effect in comparison with entirely nonspecific interactions; and, (3) at a given strength of the soft interactions, hydrophobic crowders exhibit a smaller degree of association with the protein compared to the crowders that bind nonspecifically to the protein. In light of our results, it would be very interesting to see experiments that probe the stability of a test protein in the presence of protein crowders with a variable number of hydrophobic surface amino acids.

### Funding
This work was supported by the Natural Sciences and Engineering Research Council of Canada (grant no. RGPIN-2016-05016) and was made possible by the high-performance computing resources provided by the Digital Research Alliance of Canada. The funders had no role in study design, data collection and analysis, decision to publish, or preparation of the manuscript.

### Grant Disclosures
The following grant information was disclosed by the authors:
The Natural Sciences and Engineering Research Council of Canada: RGPIN-2016-05016.
The Digital Research Alliance of Canada.

### Competing Interests
The authors declare there are no competing interests.

### Author Contributions
- Saman Bazmi conceived and designed the experiments, performed the experiments, analyzed the data, performed the computation work, prepared figures and/or tables, authored or reviewed drafts of the article, and approved the final draft.
- Stefan Wallin conceived and designed the experiments, performed the experiments, analyzed the data, performed the computation work, prepared figures and/or tables, authored or reviewed drafts of the article, and approved the final draft.

## Data Availability

The code for the Cbeta model is available at Github: https://github.com/jswallin/Cbeta-model.

## Supplemental Information

Supplemental information for this article can be found online at http://dx.doi.org/10.7717/peerj-pchem.31#supplemental-information.

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
