# Peer review of "Comparing effects of attractive interactions in crowded systems: nonspecific, hydrophobic, and hydrogen bond interactions"

_PeerJ Physical Chemistry, doi:10.7717/peerj-pchem.31_

## Round 0.1 · original submission · Major Revisions

Dear Dr. Wallin:

Thank you for submitting your manuscript to PeerJ Physical Chemistry. It has been examined by two expert reviewers whose comments are enclosed. On the basis of these reviewers’ comments and my own reading of your manuscript I have decided that your manuscript will likely be publishable after revisions.

The reviewers raise several questions about the rationale for the conditions selected, presentation of results, and comparison of results between environments that would benefit from clarification and additional discussion. I do not believe extra experiments are needed, but more detailed comparison and discussion would help clarify the work.

Reviewer two commented on the brevity of the conclusions. Expanding the conclusions would help provide perspective on unresolved questions, gaps in knowledge, and possible future directions. In particular, I found reviewer one’s comment on the value of this work in providing “important guidance for experiments of crowding effects” as a good suggestion for improving the conclusions.

Please respond to the reviewers’ comments point-by-point and revise your manuscript accordingly. It has been a pleasure working with you on this manuscript, and we wish you success in your research.
Please let us know if you have any questions.

With sincere regards,

Dr. Gerrick Lindberg

Reviewer 1 ·

Basic reporting

no comment

Experimental design

no comment

Validity of the findings

no comment

Additional comments

Bazmi and Wallin investigated the effects of crowders on protein stability via a coarse-grained protein model. Three types of crowders have been studied: volume exclusion, non-specific, and hydrophobic crowders. It is found that, for the same interaction strength and concentration, hydrophobic crowders have higher destabilizing effects compared to non-specific crowders. This is likely due to the difference in destabilizing mechanisms: non-specific interactions prefer U-type proteins over N-type proteins, while hydrophobic interactions are hindered in N-type and cause destabilization even with weakly favorable interactions with U-type. Moreover, the effects of short polypeptides as crowders are also investigated, where the balance between soft and hydrophobic interactions determines the net effects of destabilization.
Overall, this work provides interesting results regarding the effects and interplay among different types of crowders. Meanwhile, it offers important guidance for experiments of crowding effects in the stability of a protein's native state, which would be of interest to a broad audience working with proteins and peptides.
I believe this paper would be suitable for publication in PeerJ after the authors address the following comments. No further revision is needed.
Comments:
1. In Figure 6A of peptide 1, the curves seem to diverge slightly for different Npep at higher temperatures. However, in Figure 6B of peptide 2, the curves are more converged for different Npep at higher temperatures. What could be the reason for this difference between peptide 1 and 2 at higher temperatures?
2. Peptide 1 was tested for Npep ranging from 14 to 56, but peptide 2 was tested from 7 to 28. The reasons for choosing different polypeptide lengths are unclear. Authors should clarify the selection of different Npep ranges for the two peptide types.
3. In this work, the protein structure is fixed at standard values for U and N, but the secondary structure of proteins is dynamic in the cellular environment and may deviate from a perfect U/N type. Thus, it is important to discuss the crowding effects for less standard protein structures. For example, how would the hydrophobic/non-specific interactions change for non-standard protein structures?
4. Furthermore, soft interactions of proteins, such as hydrogen bonds with polypeptides, are likely to alter protein secondary structures. Therefore, the temperature dependence of destabilization and the balance between soft and hydrophobic interactions could shift accordingly. How would the alteration of protein structure affect the crowding effects of non-specific and hydrophobic interactions, as well as the interplay between the two? Some comments on the alterations of protein secondary structure upon crowding would further strengthen the relevance of this work to the cellular environment and protein experiments.

Reviewer 2 ·

Basic reporting

The language used is clear and unambiguous for the most part. The introduction is well researched and clearly presents the problem and the need for this study. The raw data has been supplied in the supplemental information. The structure mostly conforms to the journal standards except for the conclusion. The conclusion is too short and does not really summarize the major findings or conclude much. I understand that most of this was covered in the discussion, but a more in-depth conclusion would still be helpful for the reader.

The biggest issues with this paper are with figures 1,2, 5, and 6.

My issues here are ranked in terms of importance:
• Both figures 2 and 6 six compare difference between different systems but the data is not on the same page. Additionally, most of the difference in the stability are seen are low temperature. For figure 2, this is mentioned in lines 217-18 as “clear variations” but they aren’t clear form the actual figure. I have overlaid the figures myself and I can see a difference between the low temperature excluded volume and soft interaction systems, but it is hard to distinguish between the hydrophobic and nonspecific. I know some of this addressed in figure 3, but it should be addressed specifically in discussion of this figure. Additionally, the lines and symbols are too thick, which makes it difficult to see any difference between the curves.

• Figure 5A and 5D do not seem to be referred to in the text. I can seem some mention of 5A in 267 and 5D in 273-275, but there is no specific reference. It is also confusing that 5B is discussed before the discussion of 5A. Additionally, 5C is very hard to read. The black lines make it very difficult to follow which curve is which. The vertical axis seems too large for the data being presented which makes it hard to distinguish differences between the curves. Lastly, I can’t see where the 0.75 attraction strength curve intersects with the crossover line in 5C. It's not clear to me where the crossover temperature in 5D came from for this value. It might be a misreading on my part, but I am a bit confused by this.

• I am not sure why the range shown in figure 1 is necessary. It seems like the well depth is the key point of the figure. I see that the shaded gray area refers to the potential function, but the shaded area is not explicitly addressed in the text and I don’t think it is needed here.

Experimental design

This is original and important work that should be published in this journal. The research question is well defined and an important question for understanding the stability of protein structure in crowded environments such as those found in the cell. The methods are well described except for one note:

• The authors do not state whether they used package software to run their simulations or do their analysis. If they did write their own code for the simulations of analysis, that should be stated explicitly in addition to whether this code is publicly available in a repository or available by request.

Validity of the findings

All the underlying data has been provided along with a descriptive readme file. The conclusions of this work are supported by the data presented in the paper. The conclusions are limited to the work presented here and do not overextend past what was found in the study.

Additional comments

The authors have used a CG protein model to understand the destabilizing effects of soft interactions versus destabilizing effect of excluded volume effects in crowded protein systems. This type of study is very important because most proteins studies occur in dilute conditions even though cellular environments are crowded. Being able to meaningfully understand how different type of interactions affect the stability of native protein structures in the cell is therefore a very important problem that this paper addresses.

The authors come to some very interesting conclusions regarding their findings regarding the balancing of excluded volume effects and attractive interactions. The way they delineate the role of hydrophobic destabilizing effects as opposed to nonspecific interactions is especially interesting. The finding that the specificity of hydrophobic interactions with respect to the unfolded state is more important than the total number of interactions is a very exciting result and fits very well with the model proposed in figure 8.

Annotated reviews are not available for download in order to protect the identity of reviewers who chose to remain anonymous.

---

## Round 0.2 · accepted · Accept

Dear Dr. Wallin:

Thank you for submitting your revised manuscript to PeerJ Physical Chemistry. I invited one of the reviewers to reexamine it and they agree with me that the revised manuscript has addressed their concerns and is mostly ready for publication. I agree with the reviewer's lingering point that an explicit statement about the software used would be helpful for readers. I encourage you to make this modification. I have nevertheless decided to accept the manuscript for publication. Congratulations!

With sincere regards,

Dr. Gerrick Lindberg

Reviewer 2 ·

Basic reporting

The authors have made changes to the figures and the text that clarifies the issues I had with the previous draft.

Experimental design

The authors explained in the rebuttal that they used in house software and provided a link to the code on github. I am not sure if the journal requires explcitly giving the github link, but I would still suggest that the authors add some text explaining that this was done using in-house software. Something like: "*Using in house software*, runs were carried out with either 8 temperatures in the171
range kBT = 0.48 − 0.68 or 10 temperatures in the range 0.40-0.70." If the journal requires a link the software, then that should also be included.

Validity of the findings

No comment.

Additional comments

Other than the very minor mention of in-house software, I think this paper should be accepted for publication.